# Nanogels: Recent Advances in Synthesis and Biomedical Applications

**DOI:** 10.3390/nano14151300

**Published:** 2024-08-01

**Authors:** Pasquale Mastella, Biagio Todaro, Stefano Luin

**Affiliations:** 1NEST Laboratory, Scuola Normale Superiore, Piazza San Silvestro 12, 56127 Pisa, Italy; 2Fondazione Pisana per la Scienza ONLUS, Via Ferruccio Giovannini 13, 56017 San Giuliano Terme, PI, Italy; 3Department of Chemical Engineering, KU Leuven, Celestijnenlaan 200F, 3001 Leuven, Belgium; biagio.todaro@kuleuven.be; 4NEST Laboratory, Istituto Nanoscienze-CNR, Piazza San Silvestro 12, 56127 Pisa, Italy

**Keywords:** nanogels, nanoparticle synthesis, hydrogels and microgels, batch chemistry, flow chemistry, microfluidics, nanomedicine, theranostics, crosslinking, drug delivery

## Abstract

In the context of advanced nanomaterials research, nanogels (NGs) have recently gained broad attention for their versatility and promising biomedical applications. To date, a significant number of NGs have been developed to meet the growing demands in various fields of biomedical research. Summarizing preparation methods, physicochemical and biological properties, and recent applications of NGs may be useful to help explore new directions for their development. This article presents a comprehensive overview of the latest NG synthesis methodologies, highlighting advances in formulation with different types of hydrophilic or amphiphilic polymers. It also underlines recent biomedical applications of NGs in drug delivery and imaging, with a short section dedicated to biosafety considerations of these innovative nanomaterials. In conclusion, this article summarizes recent innovations in NG synthesis and their numerous applications, highlighting their considerable potential in the biomedical field.

## 1. Introduction

Recent advancements in nanomedicine have led to the development of nanogels (NGs) for drug delivery, gene therapy, and nanotheranostics [1,2,3]. NGs are polymeric nanomaterials that possess the advantages of both hydrogels and nanoparticles (NPs). Indeed, both hydrogels [4] and NGs are polymeric materials with a three-dimensional crosslinked structure (at the nanometer level) able to retain large amounts of water or other fluids without dissolving; they are often biocompatible, being therefore suitable for biomedical applications. Hydrogels can vary widely in size, from a few micrometers to several centimeters, and are used in a wide range of applications, such as medical devices, cosmetics, tissue engineering, and bandages. Their structure can be molded into various shapes to suit different needs. On the other side, NGs are characterized by their sizes between a few and a few hundred nanometers, which facilitate specific interactions at the cellular and tissue level. Moreover, like NPs, the functionalization of their surfaces allows linking various moieties at high densities (because of the high surface-to-volume ratio of nanoparticles); this can be used, e.g., for targeted delivery and/or for modulating the protein corona in biofluids [5]. Due to their inherent porosity, NGs can encapsulate both hydrophilic and lipophilic payloads very efficiently, protecting them from rapid renal clearance and from degradation (e.g., by hydrolysis or enzymatic degradation) during storage or when within biofluids, therefore increasing their shelf and circulation half-lives. Finally, they can be made responsive to specific external stimuli such as pH, temperature, or specific molecules by exhibiting changes in the gel volume and water content (“swelling”), colloidal stability, mechanical strength, and/or other physical/chemical properties [6]; this enables the controlled and precise release of drugs or other theranostic molecules. For all these reasons, NGs are often applied in advanced applications, such as targeted drug delivery, biomedical imaging, and diagnostics. However, NGs share with nanoparticles a relatively low translation: e.g., despite extensive research in the last decades, only 15 nanoparticle-based pharmaceuticals for cancer treatment are currently on the market [7]. The bottlenecks in NP translation include limited scalability, poor control over reaction parameters, extensive NP polydispersity, unsatisfactory batch-to-batch reproducibility, and large volumes of chemical reagents used, including drugs. These issues affect encapsulation efficiency and release profiles, hindering optimal treatment performance [8]. Furthermore, NGs usually suffer from low mechanical and biological stability under physiological conditions, also due to physical interactions, since they are often “softer” than other kinds of NPs.

Nanogels can be composed of a variety of natural or synthetic polymers based on their specific applications and desired properties. Natural polymers include, e.g.: (i) chitosan, derived from chitin, known for its biocompatibility, biodegradability, and antimicrobial properties; (ii) alginate, extracted from algae, valued for its ability to form gels in the presence of divalent ions such as calcium; (iii) hyaluronic acid, a natural component of the extracellular matrix, chosen for its capacity to retain water very efficiently; and (iv) polysaccharides, like in natural gums [9]. Natural polymers can also be chemically modified (see [9] for some examples), as also discussed in Section 2. Among synthetic polymers, polyacrylamide is used for its ability to form strong gels and for the ease of derivatization due to the presence of the amine group, while poly(2-hydroxyethyl methacrylate) (Poly(2-HEMA)), known for its biocompatibility and transparency, is often used in contact lenses and medical devices. Poly(N-isopropylacrylamide) (PNIPAM) is a thermoresponsive polymer useful for controlled drug release, featuring a low critical solution temperature (LCST) of around 32 °C in water; it forms temperature-sensitive gels that swell or shrink in response to temperature changes. Polyvinyl alcohol (PVA), chosen for its biocompatibility and solubility in water, forms tough films and gels [10], while poly(ethylene glycol) (PEG) is used to improve biocompatibility and reduce the immunogenicity of NGs. Polyglutamic acid (PGA), a biodegradable and biocompatible polymer, is utilized for its ability to enhance the stability and drug-loading capacity of NGs [11]. Table 1 shows the advantages and disadvantages of the natural and synthetic polymers mentioned in this review. In previous reviews, NGs have been classified according to their polymeric components [12], the types of bonds involved in forming the polymer network or its organization [13,14], their stimulus-responsive capabilities [15], routes of administration [16], and applications [17].

In this review, we will primarily concentrate on the synthesis methods of NGs, emphasizing their strengths and weaknesses. We will also describe synthesis protocols based on flow chemistry, especially microfluidics, as this can help address some of the bottlenecks for NP translation in clinics, as mentioned above. In addition, we will illustrate the various applications of NGs in different fields while also providing a brief analysis of possible challenges for clinical translation (Figure 1). Focusing mostly on relevant research from the past five years, this review aims to provide a thorough examination of the topic, highlighting current challenges and offering insights for researchers to overcome them.

## 2. Batch Synthesis

In the last several years, various techniques for NG production have been developed. NGs can be made via simultaneous or sequential polymerization and crosslinking, depending on the starting ingredients. There are numerous techniques for their synthesis, such as ionic gelation, emulsion polymerization, precipitation polymerization, inverse nanoprecipitation, self-assembly, and (micro)template-assisted polymerization [18,19]. For NGs, many polymerization processes can take place in an aqueous environment, thanks to the water solubility of most of the monomers and crosslinking agents used for the formation of NGs.

Precipitation polymerization and reverse emulsion polymerization are the most widely used techniques based on simultaneous crosslinking and polymerization for creating NGs. Precipitation polymerization is a method in which soluble monomers are polymerized in a solvent in which the resulting polymer is insoluble. This process results in the formation of polymer nanoparticles or NGs through several key steps. Initially, the (multifunctional) monomers and a crosslinking agent are dissolved in a suitable solvent. Polymerization is then initiated, usually by a thermal or chemical initiator that leads to the formation of free radicals. These radicals initiate polymerization of the monomers, and the crosslinking agent creates bonds between the polymer chains, leading to the formation of a three-dimensional network. As polymerization proceeds, the growing polymer chains become insoluble in the solvent and precipitate, forming NGs. The size and structure of the NGs can be controlled by adjusting monomer concentration, amount of crosslinking agent, and polymerization conditions, such as temperature and time; in particular, controlled radical polymerization techniques, such as atom transfer radical polymerization (ATRP) and reversible addition–fragmentation chain transfer polymerization (RAFT), can slow the reaction rate to promote the formation of uniform particles [17,19,20]. For instance, Ribovski et al. [21] used precipitation polymerization to produce fluorescently tagged PNIPAM NGs, with the degree of polymer crosslinking determining the NG hardness. In another example, Kusmus et al. [19] outlined the development of versatile epoxide-functional precursor NGs via controlled crosslinking polymerization; in a subsequent post-formation step, the epoxide moieties were functionalized with amines, azides, or thiols, as well as hydrolyzed to the corresponding diols.

Nanogels can also be synthesized using suitable emulsification techniques with an oil-soluble emulsifier. Reverse emulsion polymerization, also known as water–oil emulsion polymerization, entails the dispersion of an aqueous phase containing monomers and a crosslinking agent in a continuous oil (or organic) phase immiscible with water, with the aid of a surfactant. Polymerization takes place within the dispersed water droplets, with the monomers reacting to form crosslinked polymer chains, resulting in the formation of NGs [22]. The concentration of monomers and crosslinkers, the pH of the reaction medium, the choice of surfactant, and other parameters substantially impact the size of the NGs formed during reverse emulsion polymerization. The drawbacks of this method include the use of an organic solvent as the reaction medium and the difficulty in purifying the resulting NGs due to the presence of emulsifiers and co-emulsifiers.

While the simultaneous approach is widely adopted, NGs can alternatively be synthesized through two sequential steps: polymerization and crosslinking. This sequential approach offers enhanced control over NG properties and functionalities, a topic that will be explored further in subsequent sections. In terms of post-polymerization crosslinking, NGs are categorized into chemically crosslinked or physically self-assembled types; the first type relies on covalent bonds among different polymers, while the second relies on non-covalent interactions among them. However, many NGs incorporate both chemical and physical connections in their formation. Thus, this section will elaborate on these interactions and showcase examples of NGs created through these mechanisms. In Table 2, we schematically compare the advantages and disadvantages of the two types of syntheses, which will be discussed in more detail below. These different synthesis techniques provide versatile tools for tailoring NGs with specific properties, making them highly adaptable for applications in drug delivery, diagnostics, and tissue engineering within the biomedical field.

### 2.1. Physical Methods

Physical methods for the synthesis of gels and NGs rely on spontaneous interactions and arrangements of their components, resulting in rationalized self-assembled and self-organized supramolecular structures without covalent bonds. Despite having lower mechanical strength than covalently bonded systems, the physically constructed NGs can be preferred because they do not require additional crosslinking agents or polymerization initiators, improving their safety and biocompatibility. The primary forces driving the formation of these entangled polymeric materials are often called host–guest interactions and can include hydrogen bonds, electrostatic interactions, van der Waals forces, π−π stacking (interactions among aromatic rings), and hydrophobic interactions.

#### 2.1.1. Electrostatic Interactions

NGs based on electrostatic interactions are created by the attraction between molecules with opposite charges, which drives their self-assembly and stability. These structures are typically constructed using polymers containing ionizable or ionic functional groups, such as carboxylates, amines, or quaternary ammonium ions. Polymers commonly used to create NGs through electrostatic interactions include chitosan [23], alginate [24], hyaluronic acid (HA) [25], and poly(glutamic acid) (PGA) [26]. The synthesis of NGs by physical crosslinking involves a simple and efficient procedure, usually conducted in an aqueous environment where the polymer chains are completely solvated and initially not strongly interacting. When conditions are changed, e.g., by changing temperature, pH, or ionic strength, the polymer chains begin to interact and form crosslinks. For example, pH adjustment can protonate or deprotonate certain polymer functional groups, causing them to attract each other and form ionic crosslinks. Another type of synthesis relying on electrostatic interaction is ionic gelation, based on the use of polyelectrolytes forming crosslinks in the presence of ions; it is a simple and rapid synthesis, but it is difficult to control given the very rapid assembly of polymers and crosslinkers, causing inter-batch variations in physicochemical properties such as particle size, polydispersity, surface charge, and drug release profiles.

In general, electrostatic interactions cause the polymer chains to organize themselves into a nanoscale gel structure, effectively trapping water within the network. This straightforwardness not only simplifies production but also makes the resulting NGs ideal structures for biomedical applications, with reduced risk of toxic effects and an improved biological safety profile due to their synthesis without chemical crosslinking, which often requires non-biocompatible crosslinkers. One of the remarkable advantages of physically crosslinked NGs is their adjustable size, precisely controlled by changing parameters such as polymer concentration, ionic strength, temperature, and pH during synthesis [27]. This tuning is crucial for adapting the properties of NGs to the requirements of specific applications, such as drug delivery or tissue engineering. Furthermore, the ability to modulate the size of NGs allows for optimizing their pharmacokinetic profiles, cellular uptake, and in vivo biodistribution. The electrostatically driven formation of NGs allows the precise encapsulation of charged or polar drugs or biomolecules, which interact with the charged groups in the NG components. This versatile system finds applications in encapsulating chemotherapeutic agents for improved treatment efficacy [28] and more generally, in theranostics [29], where nanoparticles are formulated for simultaneous therapy and diagnostics. An example of the synthesis of a drug-containing NG where electrostatic self-assembly is involved is reported in Figure 2.

#### 2.1.2. Hydrophobic Interactions in Amphiphilic Nanogels

Polymers composed of hydrophobic and hydrophilic parts are called amphiphilic and can be used in the generation of amphiphilic NGs. The synthesis of amphiphilic NGs involves the dissolution of amphiphilic polymers in an aqueous solution, in which the hydrophilic segments preferentially interact with water, and the hydrophobic segments tend to avoid it. By changing environmental conditions, such as temperature, pH, or ionic strength, the polymers self-assemble, with the hydrophobic parts aggregating. Possible structures for these kinds of nanoparticles are a hydrophobic core surrounded by a shell of hydrophilic segments or more hydrophobic nanodomains in a hydrophilic matrix. Amphiphilic NGs possess the capacity to swell in aqueous and organic media due to the mixture of hydrophilic and hydrophobic compounds. However, they exhibit a lower propensity to swell in water than NGs consisting exclusively of hydrophilic polymers, ensuring, therefore, superior mechanical properties. In addition, they offer greater thermophysical and structural stability. In this formulation, the fundamental characteristics of the colloidal structure are driven by the primary structure of the polymer, including its composition, molecular weight, and branching. The type and abundance of hydrophobic groups along the polymer chains, which act as physical bonds, are particularly important, thus defining the network formation in the NG structure [30]. Moreover, this formulation facilitates the encapsulation of poorly water-soluble payloads; the cargo-loading capacity and release kinetics are set by the interaction between the hydrophobic cargo and the hydrophobic nanodomains [31]. In addition, the flexible hydrophilic matrix allows for control of the mechanical properties of the NGs and reduces the potential toxic effects of the hydrophobic groups. The control of these interactions can be achieved by varying the type and content of hydrophobic side groups on the polymer. This allows fine-tuned regulation over the loading and release profiles [32]. Consequently, the ability to finely tune the hydrophobicity of the NGs, or the hydrophilic/hydrophobic balance, will open new therapeutic options for various administration routes [33]. Furthermore, the amphiphilic structure offers the possibility to regulate and tailor their interactions with biological systems; for example, Bewersdorff et al. produced amphiphilic NGs with more or less idrophobic groups on their surface, and they were, therefore, able to control the protein corona formation and modulate interactions with biological barriers [34].

Even if the self-assembly strategy offers considerable versatility, counting exclusively on physical bonds can be restrictive. To overcome this limitation, reactive groups can be incorporated into amphiphilic copolymers, facilitating covalent crosslinking after the self-assembly process (Figure 3). This method stabilizes the properties of the nanoparticles, resulting in functional amphiphilic NGs whose network characteristics can be precisely controlled by a combination of hydrophobic physical interactions and covalent crosslinks. Covalent crosslinking of these self-assembled systems can be achieved through two main synthetic approaches. One incorporates all the necessary reactive groups directly into the amphiphilic copolymer. The other involves the reaction of the reactive copolymers with (bi)functional crosslinkers. These strategies, which employ crosslinking agents and chemical synthesis, will be discussed in detail in the following sections.

### 2.2. Chemical Methods

Covalent interactions in the synthesis of NGs can provide essential stability and functional properties. Various chemical reactions can be used to form these bonds, including free radical polymerization, click chemistry, disulfide bond formation, and carboxyl-amine reactions. Each of these reactions offers distinct advantages in terms of specificity, efficiency, and stability, helping to ensure the integrity and functionality of the network under physiological conditions.

#### 2.2.1. Covalent Crosslinking Reaction

Nanogels formulated using covalent crosslinking have stable chemical bonds between polymer chains, forming a three-dimensional network that can retain water without dissolving. By selecting appropriate polymers and crosslinking agents, the properties of the NGs can be tailored for specific functionalities, such as pH sensitivity, temperature responsiveness, or redox responsiveness. The incorporation of various functional groups during the crosslinking process also enhances the versatility of chemically crosslinked NGs, making them suitable platforms for a wide range of biomedical applications. Both the choice of the starting materials and the particle formulation methods (e.g., microemulsion [36], precipitation) can be tailored to effectively shape and stabilize the NG three-dimensional structure. In a study by Tian et al. [37], the initial fabrication of an NG exploited a modified emulsion crosslinking technique: poly(ethylene glycol) diglycidyl ether acted as a long binding agent, linking hyaluronic acid (HA) chains within the emulsions to create a loose NG structure. Next, cystamine was introduced as an additional binding agent between HA chains via an EDC/NHS (1-ethyl-3-(3-dimethylaminopropyl)carbodiimide/N-hydroxysuccinimide) reaction to create a more compact NG. In this way, the NGs became reactive to the glutathione (GSH), more abundant in a tumor environment and in particular in cell cytoplasm; GSH breaks the disulfide bridge in the cystamine, reverting the NGs to a loose structure with consequent release of the drug loaded within.

The formation of amide bonds used, e.g., for the reaction with cystamine mentioned above, is a common approach to chemical crosslinking; these bonds are typically formed from carboxyl and amine groups using carbodiimide chemistry or by generating activated esters. In the synthesis of NGs, amide bonds can serve a dual role. They can act as crosslinking points within the NG network, contributing to its structural integrity. Moreover, they can function as linkers that facilitate the conjugation of drug molecules or other bioactive compounds to the NG structure. Apart from amide bonds, numerous other chemical bonds can be used for crosslinking in NG synthesis. These include disulfide bonds, which will be discussed later, and ester bonds formed by hydroxyl and carboxyl groups.

#### 2.2.2. Click Chemistry

Click chemistry (Figure 4) involves reactions commonly used for joining two molecular entities of choice with very high chemical yield towards a single product; click reactions are widely insensitive towards solvent parameters, oxygen, and possibly water, and happen in mild (e.g., physiological) conditions [38]. Typically, these reactions can occur in an aqueous environment, ensuring compatibility with sensitive biological components. Furthermore, when applicable, these reactions evolve according to a mechanism called regiospecificity, whereby one of the possible functional isomers is generated preferentially, if not exclusively. Click chemistry is widely used in NG synthesis due to its efficiency, specificity, and ability to form stable covalent bonds. This approach has significant advantages in NG production, including short reaction times, higher productivity, and improved purity. These characteristics make click chemistry an ideal method for tailoring NGs with precise control over their properties. Reactions such as azide–alkyne cycloaddition and thiol-ene reactions (Figure 4) are particularly useful for crosslinking polymer networks within NGs. Azide–alkynic cycloaddition, which includes variants such as Cu(I)-catalyzed (CuAAC) and strain-promoted (SPAAC) reactions, involves the covalent binding of azide and alkynic groups to create 1,2,3-triazole bonds [38] (Figure 4). CuAAC offers high specificity and fast kinetics under mild conditions; however, the application of CuAAC is somewhat limited by the potential cytotoxicity of copper ions and their ability to generate reactive oxygen species (ROS), which can damage biomolecules. Sometimes, it is possible to mitigate these effects by trying to thoroughly remove the copper ions upon purification of the products. Duro-Castano et al. optimized CuAAC coupling conditions in aqueous solution prior to the preparation of polyglutamic acid-based NGs to achieve higher crosslinking efficiency using the minimum amount of catalyst. Furthermore, they established washing protocols that used acidic conditions to protonate the carboxylic acid groups and thus hinder their complexation with the remaining copper ions [39] (Figure 5). Copper-free click reactions, including SPAAC, which utilize strained alkenes like dibenzocyclooctyne (DBCO), generally involve bioorthogonal cycloadditions, which allows them to take place within living systems without interfering with native biochemical processes and are, therefore, widely applied in NG synthesis due to their biocompatibility [40]. For example, Nagel et al. reported a peptide-crosslinked NG in which dendritic polyglycerol (dPG) modified with bicyclononyne groups (BCN) to form dPG-BCN was crosslinked by a matrix metalloproteinase (MMP)-sensitive peptide ligand modified with two azides [41]. Together with thiol-ene coupling, disulfide exchange reactions, and Michael reactions, which will be discussed in detail below, these reactions highlight their exceptional suitability for NG preparation.

#### 2.2.3. Click-like Reactions

Other reactions included in the category of “click chemistry” are thiol-click-chemistry ones (Figure 4), which cover various thiol-based reactions, such as thiol−alkene and thiol−alkyne reactions, Michael addition, and disulfide exchange. Thiol-based reactions are advantageous due to their moderate orthogonality, allowing them to proceed in the presence of diverse functional groups without interference. They are highly reactive and easy to perform, typically under mild conditions, and generally produce high yields. Their efficiency, undemanding reaction conditions, and high specificity make them especially suitable for NG synthesis and other biomedical applications. In organosulfur chemistry, the thiol-ene reaction, also called alkene hydrothiolation, involves the reaction between a thiol (R-SH) and an alkene (R_2_C = CR_2_) to produce a thioether (R-S-R—note that “R” is a generic organic group, possibly different wherever it appears). Thiol-ene additions can occur through two mechanisms: free-radical additions and Michael-catalyzed additions. Free-radical additions can be triggered by light, heat, or radical initiators, which generate thiyl radicals, while thiol-ene–Michael addition is catalyzed by a base or a nucleophile.

Thiol-ene–Michael addition is a significant member of the “click” chemistry family and has many parallels with CuAAC and SPAAC reactions. However, a crucial difference is the natural presence of reactive groups. Azides and alkynes are absent in native biomolecules, allowing CuAAC to effectively functionalize only the particles in complex environments like the ones in living matter. In contrast, thiol-ene chemistry is very advantageous for conjugating or functionalizing colloids with biomacromolecules, such as proteins, due to the presence of thiols in cysteine-containing ones. It has been used for the complete crosslinking of NGs during their synthesis, but, to a reduced extent, also for their surface functionalization. These functionalized NGs can then be employed in various fields, including biosensing, bioimaging, drug delivery, and theranostics [42,43,44]. Among the most efficient Michael-type additions are the reactions between thiols and maleimides. The primary driving forces for this reaction are the electron-withdrawing effect of the two adjacent activating carbonyl groups and the release of ring strain upon product formation. The reaction between maleimides and thiol-containing biological molecules has been employed since 1949 [45]. However, it was not until 1980 that thiol–maleimide reactions were recognized as potential tools for the functionalization of nanocarriers. The maleimide–thiol reaction is widely used in functionalization protocols due to the high reactivity of maleimides under mild conditions, their selectivity for thiol groups at physiological pH, and the stability of the resulting thioether bond under physiological conditions. In addition to surface functionalization applications, the thiol–maleimide reaction is also used in the synthesis and modification of NGs for various biomedical purposes [46,47] (Figure 5). E.g., Altinbasak et al. prepared an NG system crosslinked through the Michael thiol–maleimide addition reaction, which can be degraded in a reducing environment through a thiol–disulfide exchange reaction [48]. A significant disadvantage in this reaction is the potential hydrolysis of maleimide groups to maleamic acids in aqueous solutions, which inhibits the reaction with thiols. This secondary reaction can significantly reduce the degree of functionalization and negatively affect the properties of the resulting system overall. Disulfide crosslinking, a reaction similar to the other ones discussed so far, is also a common method for NG synthesis that offers unique advantages for various applications. In this approach, the three-dimensional network structure results from the formation of disulfide bonds (-S-S-) between polymer chains having thiol moieties. In addition, the cleavable nature of disulfide bonds in response to certain stimuli, such as glutathione or reactive oxygen species (ROS) [49,50], enables the controlled release of encapsulated therapeutic agents within target cells or tissues. This responsiveness to environmental stimuli enhances the therapeutic efficacy and biocompatibility of disulfide-crosslinked NGs, making them promising candidates for biomedical applications [51,52,53].

**Figure 5 nanomaterials-14-01300-f005:**
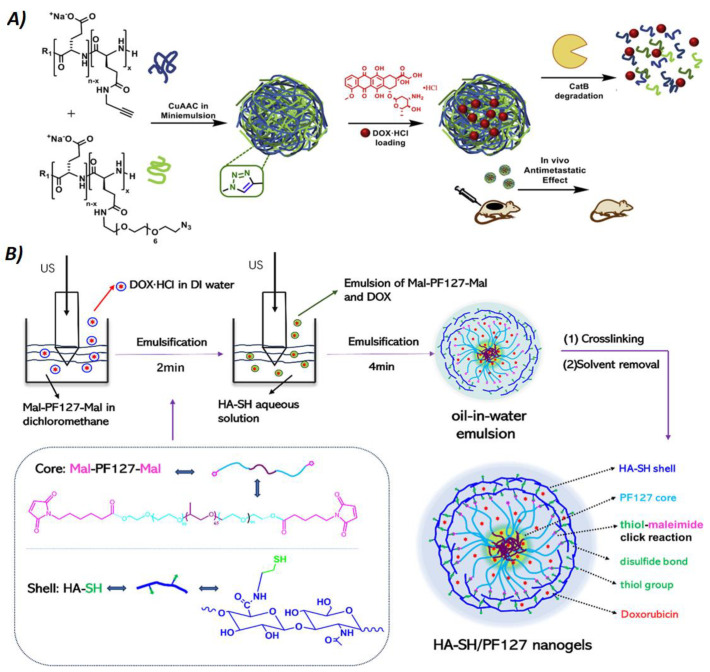
(**A**) Synthesis and mode of action of bioresponsive polyglutamic acid nanogels (NGs): NGs are obtained by miniemulsion of azide- and alkyne-modified polyglutamic acid chains (blue and green filaments, respectively) crosslinked by click reactions (CuAAC) and then loaded with doxorubicin (DOX). The image also shows the degradation mechanism mediated by Cathepsin B (CatB), a lysosomal enzyme that is overexpressed in the stroma of some types of tumors. Reprinted with permission from Ref. [39]. (**B**) Illustration of the preparation of mucoadhesive hyaluronic acid thiol (HA-SH)/pluronic acid (PF127) NGs by thiol–maleimide click reactions. Using an ultrasonicator probe (US), NGs were produced by a double emulsion technique: first by sonicating maleimide-modified PF127 with doxorubicin and then by adding in the second emulsion hyaluronic acid modified with thiols. Reproduced from Ref. [47] with permission from copyright © 2024, American Chemical Society.

## 3. Flow Chemistry Synthesis

The synthesis routes described above can also be used in flow chemistry. In this approach, also known as continuous processing or continuous flow chemistry, two or more streams of different reagents are pumped at specific flow rates into a chamber, tube, or microreactor. A reaction occurs, and the flow containing the resulting compound is collected at the outlet. To generate the final product, the solution can also be directed to subsequent loops of continuous reactors [54,55,56]. This route offers significant benefits over traditional batch chemistry, including enhanced mass and heat transfer, improved safety and reproducibility, increased reaction efficiency, reduced waste, and better scalability [57] (Table 3). Due to the intrinsic characteristics of continuous-flow reactors, it is possible to exploit reaction conditions not achievable in batch processes, allowing for precise control and real-time monitoring and resulting in high-quality products and streamlined processes. Advancements in 3D-printing flow setups and affordable electronic toolkits have made flow chemistry more accessible [58]. Continuous manufacturing, often coupled with photochemistry and photocatalysis, improves performance and safety while reducing costs [59,60,61]. Key challenges include solvent compatibilities and byproduct formation, necessitating inline analysis and purification strategies. Optimizing mass and heat transfer is crucial in flow chemistry; efficient mass transfer correlates with mixing efficiency, while heat transfer can be optimized by a high surface-to-volume ratio and by large heat-exchange surfaces in microchannels. Efficient heat transfer allows for isothermal or superheated conditions, improving chemical selectivity and safety. Small reactor volumes and precise reaction conditions provide, besides better mass and heat transfer, increased safety; some hazardous or impractical reactions can be safely conducted in flow conditions [62]. Flow chemistry allows for using lower reagent amounts, reducing costs and environmental impact. Scale-up can be achieved through numbering up or sizing up, maintaining the benefits of microreactor environments [63,64].

### 3.1. Microfluidics Systems

Several flow chemistry-based devices have been developed and optimized, and among them, microfluidics stands out as one of the most promising ones. Microfluidics technology is based on fluidic circuits where the channels have lateral dimensions of ten to hundreds of micrometers and aims at miniaturizing the manufacturing nanoparticle production apparatus (Figure 6). Compared to bulk synthesis, this approach leads to narrower size distributions, higher encapsulation efficiencies, sustained drug release profiles, and highly controlled synthesis conditions. Moreover, in the photochemistry case, the small lateral dimensions of the microchannels allow a better and more homogeneous illumination, even when molecules with high extinction coefficients are present. Additional benefits of microfluidics include low variation between batches, high throughput, reduced reagent volumes, low instrumental footprint, and scalable production yields. For these reasons, microfluidics is deemed a promising approach for the design of advanced micro- and nano-particles and micro- and nano-gels [65]. Microfluidic approaches can be employed for the production of several types of nanoparticles, from polymeric to lipidic to inorganic ones, by changing the microfluidic setup and mixing parameters [66]. E.g., for polymeric nanoparticles in general, several production strategies can be used to exploit microfluidics, like (i) nanoprecipitation, with the mixing time between solvent and non-solvent phases more easily tweakable for better control of nanoparticle nucleation and growth; (ii) self-assembly, which leverages the pH- or temperature-responsive behavior of amphiphilic derivatives; and (iii) droplet-based methods, which adopts immiscible phases to create flowing micro-emulsion droplets that act as micro-reactors. While nanoprecipitation and self-assembly methods are prone to channel clogging due to undesired interactions among reagents, droplet-based approaches provide rapid heat and mass transfer, enabling faster reaction kinetics and precise control over droplet size and composition [67].

Common microfluidic designs are based on hydrodynamic flow focusing (HFF) and staggered herringbone micromixer (SHM) [66]. HFF exploits the laminar flow regime typical of microfluidic platforms: a narrow stream of NP precursors (e.g., lipids or polymers dissolved in a solvent) flows parallel to an antisolvent (e.g., water or buffer) from two side channels. In this case, mixing occurs through diffusion because of the laminar flow [68]. In contrast, SHM induces micromixing through chaotic advection caused by structures on the microchannel floor or walls, achieving shorter mixing times at lower flow rates and, therefore, reducing dilution and synthesis times [69,70]. Such passive mixing technologies can be coupled with an active methodology to reduce the mixing time, control the nanoparticle size, and increase the throughput; active micromixing utilizes external energy sources to enhance mixing by disrupting the laminar flow regime, leading to faster homogenization and producing smaller NPs with narrower size distributions [71,72]. In this context, ultrasound is noteworthy, as it is generally employed to boost nucleation rates and significantly reduce NP mean size and polydispersity compared to other passive syntheses [73,74,75].

However, the employment of microfluidics faces several challenges. A possible difficulty arises from the fact that many materials used for microfluidic devices, like polydimethylsiloxane (PDMS), swell upon contact with organic solvents, potentially adsorbing small molecules and affecting device structure and nanoparticle production efficiency [76]. Moreover, the strong dependence of nanoparticle properties on multiple experimental factors calls for the application of a proper design of experiments (DoE). DoE, and in particular response surface methodology (RSM), is meant to identify critical variables and their interactions in order to predict the most suitable experimental conditions while minimizing costs and reagent waste [77,78,79].

### 3.2. Nanogels Produced through Microfluidics

Various methods for producing NGs via microfluidic platforms have been explored [17,20,80]: (i) Chemical gelation, involving emulsifying low-viscosity monomer solutions, followed by photoinduced crosslinking and polymerization, resulting in mechanically strong microgels and NGs; however, without proper optimization, these particles may not be easily enzymatically cleaved or metabolized, and photo-irradiation can be harmful to encapsulated biological species. (ii) Gelation induced by temperature changes after emulsification: e.g., heated agarose solutions can be cooled to form micro- or nano-gels. This approach has challenges in maintaining temperature differences across the microfluidic device and can potentially damage bioactive species. (iii) Coalescence-induced gelation: this strategy involves the coalescence of biopolymer droplets with crosslinking agent droplets, as demonstrated with alginate hydrogel microbeads; the productivity depends on droplet collision probabilities. (iv) Reversible shear thinning: droplets are formed from shear-thinning polymers that restore their network structure after emulsification. (v) Internal gelation: droplets contain a gelling polymer and a bound crosslinking agent. A compound in the continuous phase diffuses into the droplets, releasing the crosslinking agent and causing gelation; (vi) external gelation: droplets of gelling polymer are emulsified in a continuous phase containing a crosslinking agent. Diffusion of the agent into the droplets causes gelation.

These kinds of techniques were initially used to produce microgels, and some of the optimizations carried out in those cases and the applications of those microgels can also be considered for NGs. E.g., Zhao et al. [81] developed an injectable scaffold using droplet-based microfluidic technology and photo-crosslinking to synthesize bone marrow stromal cell (BMSC)-laden microgels; with a similar technique, Feng et al. [82] engineered multifunctional microgels with a precursor suspension containing kartogenin-loaded cyclodextrin nanoparticles (KGN@CD NPs), BMSCs, gelatin methacryloyl (GelMA), and phenylboronic acid-grafted methacrylate hyaluronic acid (HAMA-PBA). These microgels were then assembled using dynamic crosslinking between dopamine-modified hyaluronic acid (HA-DA) and phenylboronic acid groups on the surface of the microspheres. Upon injection into a cartilage defect, HA-DA facilitated adhesion to native tissue, and the microporous microgel assembly, along with sustained KGN release, promoted BMSC chondrogenesis and cartilage repair [82]. Seiffert and Weitz [83] used microfluidic devices to produce monodisperse poly(N-isopropylacrylamide) (pNIPAAm) microgels via a polymer-analogous crosslinking reaction, achieving higher crosslinking efficiency and greater homogeneity compared to classical free-radical crosslinking copolymerization techniques. Zhang et al. [65] prepared droplet-formed sodium alginate biomicrogels (hydrogel microbeads derived from biopolymers), whose synthesis is usually done in two stages: emulsification followed by gelation of the resulting droplets by chemical or physical crosslinking of the biopolymer. They generated stable alginate NGs through Ca^2+^-mediated crosslinking after having compared external and internal gelation in their microfluidic preparation. They showed that internal gelation had a limited application in this field because it did not allow control over morphology, and the resulting microgels were soft and not colloidally stable. In contrast, external gelation produced stable microgels with good control of the structure [65].

Microfluidic synthesis of NGs has demonstrated significant improvements in control and efficiency [20,80,84,85,86]. E.g., Bazban Shotorbani et al. [87] synthesized alginate NGs via ionic gelation using hydrodynamic flow-focusing microchips, achieving precise size control and monodispersity. Mahmoudi et al. [88] fine-tuned the flow rate ratio on microchips to form alginate NGs, while Huang et al. [89] created hyaluronic acid NGs via photo-click crosslinking on a microchip platform. Majedi et al. [90] utilized modified chitosan on a T-shaped microfluidic chip, and Chiesa et al. [91] formed chitosan/sodium triphosphate (TPP) NGs using a staggered herringbone micromixer.

Pessoa et al. [92] used microfluidic microchips in order to attempt to mitigate fouling issues in the production of chitosan/ATP nanoparticles, while Whiteley and Ho [93,94] addressed these fouling challenges by creating a 3D flow-focusing profile on a coaxial flow reactor (CFR), producing nanoparticles with smaller size and better monodispersity than 2D methods. They also tried to predict the interaction effects of process factors (component concentrations and flow ratio) on the size, PDI, and encapsulation efficiency of the NGs [93,94]. Giannitelli et al. [1] synthesized hyaluronic acid (HA) and linear polyethyleneimine (LPEI)-based NGs for controlled doxorubicin (DOX) delivery using a pressure-actuated microfluidic chip. The formation of NGs was confirmed by nuclear magnetic resonance (NMR) and Fourier-transform infrared (FTIR) analyses. NG specimens were also evaluated in terms of size and morphology through dynamic light scattering (DLS), atomic force microscopy (AFM), scanning electron microscopy (SEM), and tunneling electron microscopy (TEM) analyses, in order to define a correlation between the active tuning of the flow focusing geometry and the physical features of the resulting nanoscaffolds. The optimized DOX-containing NGs demonstrated significant antitumor and anti-metastatic effects in vivo [1].

## 4. Stimuli-Responsive Nanogels

Nanogels for biomedical applications require stability in the environments of application but can also be engineered to adapt and respond to external signals by altering their chemical and physical properties. NGs can respond to a variety of stimuli, including physical (temperature, light, ultrasound, magnetic/electric fields, pressure), chemical (pH, redox potential), and biological (enzymes, specific biomolecules) ones. Stimulus-responsive NGs are promising nanoformulations that, by their selective response to environmental signals, improve therapeutic precision, offering versatile platforms for targeted therapies and diagnostics and minimizing systemic toxicity. Continued research into optimizing these nanomaterials has the potential to realize revolutionizing biomedical technologies while also advancing precision medicine and patient care. Other reviews have described in more detail the stimuli-response properties of NGs [15,95,96], so only a brief overview is provided below.

Among the various stimuli, pH is one of the most commonly used in biomedical science. Healthy tissues have a pH around 7.4, and tumor tissues range from 6.5 to 7.0; within cells, the cytosol has a pH similar to blood (7.4), the Golgi apparatus is at about 6.4, endosomes range from 5.5 to 6.0, and lysosomes are the most acidic at 4.5 to 5.0. pH-reactive NGs exploit these variations to deliver drugs in a targeted manner, minimizing off-target drug loss. These NGs are typically synthesized by incorporating acidic [97] or basic functional groups into the polymer backbone or using crosslinking molecules that degrade under specific pH conditions [98]. This design enhances the swelling and degradation of the NG, promoting the release of the encapsulated cargo [99].

Redox-responsive nanocarriers show great promise for delivering payloads within cells by exploiting the natural redox gradient between intracellular and extracellular environments to trigger the release of encapsulated substances. The antioxidant glutathione (GSH) primarily regulates these redox potentials, and its cytosolic concentration in cancer cells is significantly higher than in normal tissues. This difference highlights the importance of redox-responsive drug delivery systems for cancer-targeted therapies [100]. Recent studies on NGs have focused on incorporating disulfide-based crosslinking monomers to achieve precise control over degradation kinetics. This strategy optimizes the balance between crosslinking and the encapsulation/release of therapeutic agents, leading to the development of NGs specifically aimed at intracellular release in antitumoral applications [48,101,102,103].

Magnetic-responsive NGs combine magnetic properties with the ability to respond to external stimuli [104]. These NGs consist of magnetic nanoparticles incorporated into a polymer gel matrix that can react to various stimuli, such as temperature and pH changes, as well as magnetic fields. The unique physical properties of these NGs hybridized with magnetic nanoparticles offer several strategies for their use in biomedicine. For example, like non-responsive magnetic nanogels, they can be used as contrast agents in magnetic resonance imaging (MRI) or for the magnetic separation of target cells bound to NGs in aqueous media, both in vitro and in vivo. Moreover, due to both their magnetic properties and responsiveness, they are suitable for the controlled release of drugs at the tissue level [105]. E.g., after injection into the body, the drug-loaded magnetic NGs can be guided by magnetic forces to specific locations in vivo; depending on the gel’s responsiveness, these NGs can swell or shrink in response to changes in local temperature or pH, releasing the drugs in a controlled manner. In the case of thermoresponsive NGs, the hybridized magnetic nanoparticles can be used as components that generate heat in response to varying external magnetic fields or near-infrared radiation (NIR) [29,104,106].

Recent years have seen significant advances in NG technology, particularly in the development of systems that can recognize and respond to multiple stimuli. This progress involves the integration of various response functions into a single platform, thereby improving the versatility, effectiveness, and/or specificity of NG-based therapies [107,108]. By incorporating two or more stimulus-responsive components within a single NG delivery system, researchers aim to achieve a higher level of precision in therapeutic applications [109]. These sophisticated NGs can dynamically adjust their behavior in response to a combination of environmental stimuli such as pH, temperature, redox potential, and light exposure, enabling therapeutic payloads to be released in a controlled manner [110] (Figure 7).

## 5. Applications

As already mentioned, due to their unique structural and functional properties, NGs have garnered substantial interest in biomedical applications. Their biocompatibility and capacity to encapsulate drugs, proteins, and other biomolecules allow for targeted and controlled release, reducing side effects and enhancing treatment efficacy. As already mentioned, the possibility of stimuli-responsive NGs allows drugs to be delivered in a specific manner. NG cellular uptake and biodistribution behavior can also be regulated by different types of surface functionalization; factors such as polarity and surface charge affect the hydrophilicity of the NGs and their blood-circulation time. It should also be noted that the variety of synthesis methods mentioned above are not exclusive to single applications but can be adapted to formulate NGs for different purposes. In recent years, NGs have achieved considerable advancements in terms of design, optimization, functionalization, and application. Therefore, in this section, we will mainly discuss recently published biomedical applications.

### 5.1. Nanogels for Drug Delivery

Nanogel-based drug delivery systems are very efficient in precisely delivering drugs to their target sites, significantly reducing toxicity to surrounding healthy cells. This remarkable potential has led to extensive research into their application for the treatment of diseases with high morbidity and mortality rates, with the aim of improving traditional therapies and patients’ quality of life. Many NGs have a high encapsulation efficiency and drug-loading capacity and can be suitable for transporting both hydrophilic and hydrophobic drugs, including small molecules such as chemotherapeutic agents and inhibitors, as well as macromolecules such as proteins, DNA, or RNA [111,112,113]. Depending on the route of administration, NGs encounter various physiological barriers during drug delivery. This necessitates specific properties in the used NGs, such as mucoadhesivity and mucopenetrativity for mucosal routes, clearance-avoidance systems for intravenous routes, or sophisticated mechanisms capable of crossing the blood–brain barrier for delivery in the central nervous system. Several reviews [114,115,116] have examined these physiological barriers and the strategies for NGs to overcome them, illustrating typical cases, and, therefore, we will not elaborate further on this topic.

A big part of the research on NGs for drug delivery focuses primarily on cancer therapy; NGs designed for chemotherapeutic drug delivery play a crucial role in improving the efficacy and reducing the side effects of cancer treatments. These particles are designed to encapsulate chemotherapeutic agents, protecting them from degradation and improving their delivery to tumor sites [117,118]. She et al. [119] developed a hypoxia-degradable zwitterionic phosphorylcholine NG (^H^PMPC NG) using an azobenzene-based crosslinker (Figure 8). This NG degrades under hypoxic conditions, triggering the collapse of its structure and the rapid release of the drug (DOX) into tumor tissues. The NGs showed prolonged accumulation in glioblastoma tissues and effectively inhibited the growth of this highly malignant tumor. However, the heterogeneity of tumors and the underlying limitations of some anticancer drugs can lead to incomplete eradication of cancer cells when using a single compound for treatment, with possible tumor recurrence. Combining multiple chemotherapeutic agents with different mechanisms of action, i.e., using “drug cocktails”, can synergistically increase therapeutic efficacy. As a result, there has recently been an intensification of loading NGs with multiple drugs [120,121]. For example, Zhang and his colleagues [122] designed a hyaluronic acid NG that can deliver DOX due to its cationic nature. The NG crosslinker used is cisplatin, which binds to the carboxyl (-COOH) side-groups of hyaluronic acid. This crosslinking stabilizes the drug-loaded NGs, preventing premature release during blood circulation. Besides focusing on specific diseases, many studies have created NG-based vectors for various applications, such as ocular administration; this innovative, noninvasive, and safer NG-based delivery method shows great potential for treating numerous conditions [123,124].

### 5.2. Nanogels for Bioimaging

Conventional imaging methods include various techniques used to visualize internal body structures for the diagnosis, monitoring, and treatment of medical conditions. For example, computed tomography (CT) combines X-rays with computer technology to generate detailed three-dimensional images of organs, bones, and tissues, proving invaluable for visualizing anatomical details and evaluating trauma, tumors, and cardiovascular diseases. Magnetic resonance imaging (MRI) employs magnetic fields and radio waves to produce detailed images of soft tissues, brain, spinal cord, and joints, making it ideal for diagnosing neurological problems, musculoskeletal injuries, and central nervous system disorders. Despite their wide use and significant contributions to medical diagnostics, these methods have limitations. In fact, the low resolution and limited signal-to-noise ratio (SNR) of these imaging techniques limit their wider application. Therefore, researchers have developed different contrast agents, i.e., substances that improve the visualization and definition of internal body structures by increasing the contrast between normal and abnormal tissues or between different types of tissues. This allows clearer and more detailed images, aiding the diagnosis and monitoring of medical conditions. Gadolinium is a transition metal used as a contrast agent in imaging techniques such as magnetic resonance imaging (MRI). It is often bound to organic molecules to increase its solubility and stability in a biological environment and sometimes to increase its concentration in certain tissues or cells. Due to its magnetic properties, gadolinium enhances the MRI contrast between different tissues in the body, allowing more detailed and accurate images. Despite its effectiveness in improving diagnosis, it has been reported that gadolinium can accumulate in body tissues, especially in the brain, and even in patients with normal renal function, although the extent and clinical implications of this phenomenon are still being studied and discussed. Among the various nanocarriers, contrast-agent-containing NGs are particularly popular due to their high water content and their ability to encapsulate a wide range of cargo [125,126] (Figure 8). Kimura et al. [127] developed ultra-small gelatin NGs as contrast agents for MRI that cannot pass through the blood–brain barrier. They used a γ-radiation crosslinking technique for the formation of these NGs and conjugated chelating agents to the protein constituting the NGs in order to load gadolinium (Gd) and form gadolinium-coordinated gelatin NGs (GdGN). In vivo studies confirmed the safety and effectiveness of GdGN as MRI contrast agents. GdGN were quickly eliminated via renal excretion within 90 min and did not cross the brain barriers. In their study, Shi and colleagues [128] synthesized (AuNP)-loaded γ-polyglutamic acid (γ-PGA) NGs using a dual-emulsion method for CT imaging of tumors. Initially, γ-PGA was activated with 1-ethyl-3-[3-(dimethylamino)propyl] carbodiimide hydrochloride (EDC) and then emulsified. Subsequently, polyethylenimine (PEI)-coated Au NPs [(Au0)200-PEI-NH2-mPEG]), synthesized by the reduction of (HAuCl_4_) with NaBH_4,_ were crosslinked in situ. In vivo CT imaging of the tumors showed that the γ-PGA-[(Au0)200-PEI-NH2-mPEG] nanoparticles effectively accumulated within the tumors and provided a clear visualization of the site.

Multimodal imaging integrates multiple imaging techniques into a single procedure to provide complementary information about biological tissues. It combines modalities such as CT and MRI to enhance diagnostic sensitivity and specificity. This approach allows for a comprehensive assessment of anatomical structures and functional characteristics, aiding in early diagnosis, therapy monitoring, and understanding of pathophysiological processes. In the study of Sun et al. [129], PEI partially modified with polyethylene glycol (PEG) was used to encapsulate AuNPs and load gadolinium through chelation. Alginate NGs (AGs) were obtained by a double-emulsion process, where the PEI-Au-Gd particles acted as crosslinkers to crosslink the alginate by exploiting the activated carboxyl groups. The AG/PEI-Au-Gd NGs exhibited higher T1 relaxivity in MRI and greater X-ray attenuation in CT compared to PEI-AuGd nanoparticles and conventional iodinated contrast agents. Due to their enhanced cellular uptake relative to PEI-Au-Gd nanoparticles, AG/PEI-Au-Gd NGs represent a dual-mode MR/CT imaging system for improved visualization of tumor cells in vitro and enhanced imaging of tumors in vivo.

### 5.3. Nanogels for Regenerative Medicine

Nanogels have emerged as a promising tool in regenerative medicine, offering unique advantages due to their nanoscale size, high water content, and versatile structure. For this purpose, these hydrophilic polymeric networks are designed to encapsulate and deliver bioactive molecules like growth factors, cytokines, and genetic material in a controlled and targeted manner. The gel’s ability to mimic the natural extracellular matrix and provide a supportive environment for cell growth and differentiation makes it an ideal candidate for tissue engineering and regenerative therapies. This adaptability increases its potential to facilitate the repair and regeneration of damaged tissues and improve the integration of implanted biomaterials.

A field of application for these systems is bone tissue regeneration, where osteoclasts and osteoblasts handle bone resorption and formation, respectively; this is an intricate process involving other possible cell types and numerous intracellular and extracellular signaling pathways, including cytokines and growth factors [130]. However, natural physiological mechanisms are insufficient for repairing large bone defects. The preferred treatment for critical bone defects is still autologous bone grafting, known for its osteoconductive (via bone fragments), osteoinductive (via growth factors), and osteogenic (via cells) properties. Nonetheless, this method often leads to chronic pain, infections, iatrogenic fractures, and suboptimal aesthetic outcomes. Nanocarriers can replicate the natural nanostructure of bone, potentially forming a precisely controlled porous microstructure that boosts osteoconduction [131]. NG formulations can serve as injectable carriers for systemic or localized delivery of drugs or genetic material; moreover, they can be integrated into scaffolds to accurately host and release active substances during tissue growth and/or to adjust the scaffold’s physical properties [88,132].

Another important application of NGs is in wound healing. When the skin surface is damaged, the timely restoration of its integrity becomes critical. The wound-healing process takes place through stages of hemostasis, inflammation, proliferation, and remodeling. While minor wounds usually heal spontaneously in a few days, conditions such as diabetes, vascular insufficiency, or cancer can cause chronic wounds that resist normal healing. Delayed recovery of chronic wounds prolongs tissue regeneration, causing structural and functional damage. Due to their substantial water content, compatibility with natural extracellular matrices, ability to take desired shapes, and effective drug delivery capabilities, hydrogels have long been favored for wound healing. NGs inherit many of these advantages and have been designed for applications in managing bleeding and promoting wound healing [133,134,135]. For example, Zhang et al. [136] developed a novel nanocomposite consisting of chitin NGs and rectorite (a mineral with hemostasis properties) for effective hemorrhage control. Chitin chains are intercalated into the rectorite, and vigorous mechanical agitation produces chitin NGs. These NGs assemble onto rectorite nanoplates through electrostatic interactions, forming a sandwich-like structure. In experimental models, the nanocomposite achieves hemostasis in 121 s in rat tail incisions and shows superior hemostatic activity compared with Celox, a commercial chitosan-based hemostatic, in rabbit artery injury models [136]. The enhanced biocompatibility and hemostatic efficacy of the chitin/rectorite nanocomposite make it a promising and cost-effective option for hemorrhage management.

In addition to the applications mentioned above, NGs have also been studied for various other uses in regenerative medicine: cardiac repair [137,138], regeneration of ischemic limbs [139], coating of biointerfaces [140], etc. Despite the limited examples discussed in this review, extensive research in this area suggests that promising and encouraging applications of NGs in regenerative medicine are likely to emerge in the near future.

## 6. Characterization, Biocompatibility, Safety, and Long-Term Stability of Nanogels

After introducing the various applications of NGs, this chapter will address the potential challenges for their clinical translatability. Currently, there are no NG-based drugs or systems approved for clinical use. However, preclinical studies have shown promising results, indicating significant potential for their future clinical application. Current research efforts involving NG-based systems are focused on conducting in vitro and in vivo studies to validate their efficacy and safety in disease treatment. One of the biggest challenges for clinical translatability is reproducibility and scalability. As described in the chapter on synthetic methods, NG production involves a multitude of parameters and variables; different conditions and reaction settings produce NGs of varying sizes, shapes, and behaviors, posing a significant challenge in obtaining reproducible results from batch to batch [141].

However, standardized procedures for their characterization and the determination of their properties prior to use in specific applications have not yet been established. Currently, characterization techniques providing detailed information on the structure of NGs are based on established methods that have long been used for hydrogels and/or for the characterization of other polymeric nanoparticles. Some of the most used techniques are as follows:Electron and atomic force microscopy: to observe the morphology and size of particles.X-ray diffraction and NMR spectroscopy: to analyze the chemical structure.Dynamic light scattering (DLS): to determine size distribution.Thermogravimetric analysis (TGA): to assess thermal stability and to help in studying the composition, water content, and porosity of (nano)gels.

In addition, other fundamental properties, such as swelling behavior, solubility, rheology, and pore size, must be considered. The absence of a standardized characterization framework complicates the comparison of results from different studies and the evaluation of NG performance in clinical and industrial applications.

The cited inter-batch variability can affect the consistency of this performance, making it difficult, e.g., to ensure uniformity in their therapeutic efficacy and safety. Overcoming these challenges is crucial for advancing NG-based systems towards clinical application, and this could also be achieved by the use of microfluidic systems, as explained in Section 4. In any case, it is crucial to implement regulatory guidelines that oversee the synthesis and development of NGs, ensuring stringent adherence and upholding high standards of quality and safety. These guidelines would streamline the process of clinical translation, thereby facilitating the effective transition of NGs from preclinical studies to clinical applications.

Another significant challenge associated with NG systems is their biocompatibility and safety in humans; even if NGs, especially the ones composed of natural polymers, are usually considered more biocompatible than other kinds of nanoparticles [142], these features have yet to be fully explored. Indeed, NG-based carriers offer promising therapeutic potential but also present risks, so they require further research into their safety, feasibility, and long-term stability as a treatment modality. Potential immunogenicity may result from interactions between the NGs and the drug delivered into the body or from interactions with biological components, resulting in allergic, anaphylactic, or hypersensitivity reactions. Biocompatibility and safety need to be designed and described from the very first stage of in vitro product testing. Using established biocompatible polymers for the synthesis of NGs can help towards this end, as they have already been tested for safety and biocompatibility; however, extensive investigations of the potential toxicity of NGs resulting from prolonged exposure and possible accumulation in tissues remain to be performed. In general, the standard set of formal toxicological analyses conducted in the preclinical setting for any new drug (starting from cell cultures to in vivo tests) should be sufficient to detect tissue-specific adverse effects associated with a nanomedicine. This is a useful guiding principle; however, it is important to recognize that additional tests specific to the behavior of specific NG formulations may be needed in the preclinical setting. Summarizing clinical studies on NGs, it is notable that they are primarily administered through topical application, topical injection, or subcutaneous injection [143]. The intravenous injection route, commonly used in preclinical research, has not yet been widely analyzed, likely due to significant challenges in addressing safety, as well as efficacy concerns with systemic administration. Furthermore, while clinical studies involving NGs are increasing, most are currently in phase I or II trials [144,145], indicating that their clinical translation still requires substantial advancement.

## 7. Conclusions and Future Perspectives

Nanogels, first developed in the late 1990s as part of polymer science and nanotechnology [2], have undergone significant evolution, becoming an integral part of modern nanomedicine studies. This review provides a concise exploration of NGs, highlighting their synthesis techniques and some biomedical applications. NGs represent a key advance in drug delivery, potentially capable of precise targeting and controlled release.

Even though there is still a significant journey to bring NGs from bench to bedside, the results presented in the literature offer a positive outlook for NGs as a novel drug delivery system or diagnostic tools: (i) they hold promise for personalized therapies in regenerative medicine and (ii) they contribute to advances in biomedical imaging. Despite these advances, several challenges hinder their widespread clinical application. Many current synthesis processes necessitate crosslinking agents that are not entirely biocompatible, as well as extreme pH and temperature conditions, which could damage some cargo; to address these limitations, the use of peptide-based NGs (PBNs) could serve as an innovative solution [146,147,148].

Nanogels are still in the early stages of development and require extensive research to address issues such as reproducibility, scalability, efficient targeting, bioavailability, potential toxicity, and long-term stability. To facilitate their clinical translation, future research should focus on developing scalable manufacturing methods and on thoroughly investigating the benefits of NG-based therapies over traditional ones. Standardizing manufacturing processes and ensuring batch consistency are critical for regulatory approval and clinical adoption. Using flow chemistry approaches, especially based on microfluidics, could help in this aspect, but the research on quantifying the advantages of this approach over conventional ones (based on batch chemistry) is still in its infancy. Addressing biocompatibility and long-term safety through rigorous preclinical and clinical studies is also imperative. Future studies should aim to establish reliable large-scale production techniques for NGs without compromising their properties, preparing them for extensive clinical evaluation.

## Figures and Tables

**Figure 1 nanomaterials-14-01300-f001:**
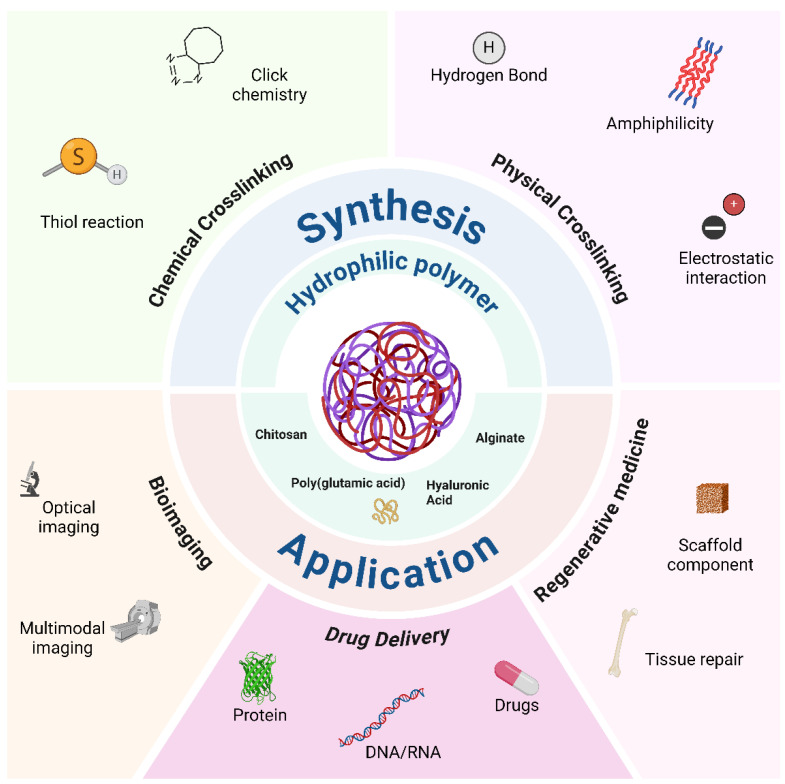
Schematic synopsis of possible nanogel synthesis routes and applications.

**Figure 2 nanomaterials-14-01300-f002:**
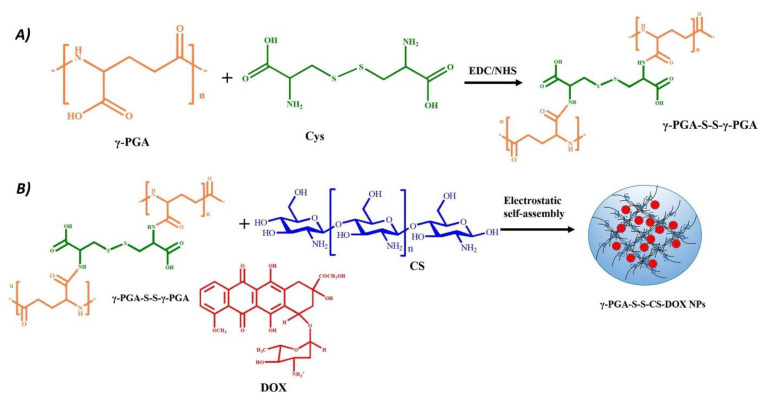
Schematic illustration of the synthesis of a nanogel (NG) using electrostatic self-assembly. (**A**) Polyglutamic acid (PGA) is covalently modified with cystine (Cys) to obtain a pH/redox-responsive NG. (**B**) The modified PGA is mixed with doxorubicin (DOX) and chitosan (CS) in an aqueous environment to formulate NGs by electrostatic self-assembly. Reprinted with permission from Ref. [26].

**Figure 3 nanomaterials-14-01300-f003:**
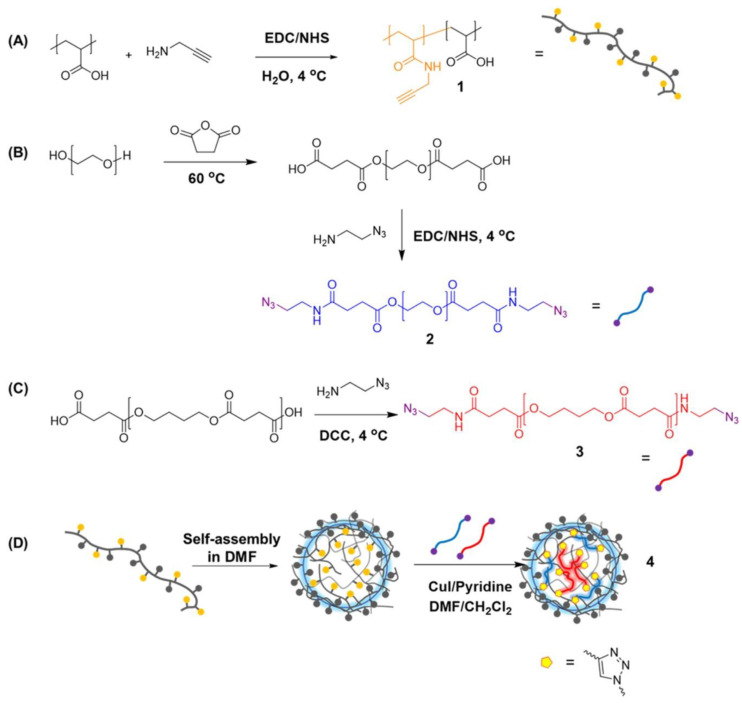
Illustration of the synthetic route for the preparation of random amphiphilic copolymers using poly(acrylic acid) (PAA) modified with alkyne groups (**A**) and their macromolecular crosslinkers poly(ethylene glycol) (PEG) (**B**) and poly(butylene succinate) (PBS) (**C**) modified with azides. Amphiphilic NGs (**D**) were created by hydrophobicity-driven self-assembly of the copolymers (**A**) and subsequent crosslinking with a mixture of hydrophilic (**B**) and hydrophobic (**C**) crosslinkers, resulting in amphiphilic NGs with biodegradable and/or water-soluble crosslinkers. Reprinted with permission from Ref. [35].

**Figure 4 nanomaterials-14-01300-f004:**
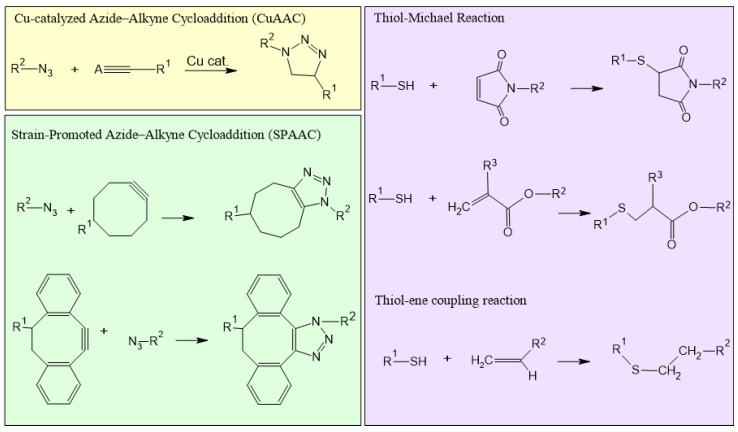
Schematic illustration of different types of click reactions. CuAAC in yellow, copper-free click reactions (SPAAC) in green, and click-like reactions in violet.

**Figure 6 nanomaterials-14-01300-f006:**
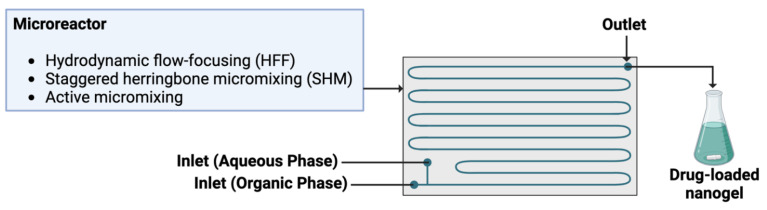
Schematic diagram of the simplest microfluidic experimental setup.

**Figure 7 nanomaterials-14-01300-f007:**
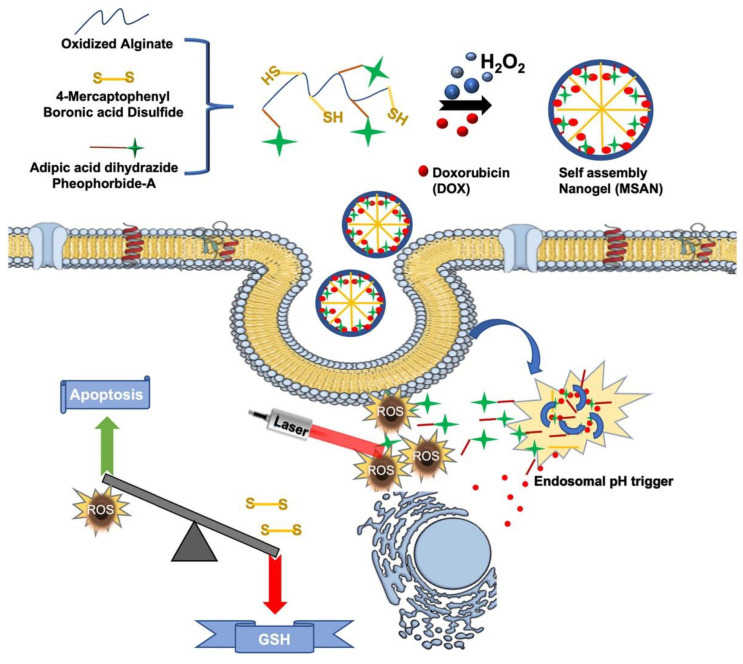
Schematic illustration of the multi-stimuli-responsive alginate nanogel (MSAN) for chemo-photodynamic therapy. Using a skeleton of oxidated alginate conjugated with 4-mercaptophenyl boronic acid and Pheophorbide-a (photosensitizing agent), the nanogels were formed by self-assembly in the presence of hydrogen peroxide (H_2_O_2_) and doxorubicin (DOX). The image also shows the mechanism of action within a cell with particle degradation triggered by acidic pH conditions and/or assisted by phototherapy. ROS: reactive oxygen species; GSH: glutathione. Reprinted with permission from Ref. [110].

**Figure 8 nanomaterials-14-01300-f008:**
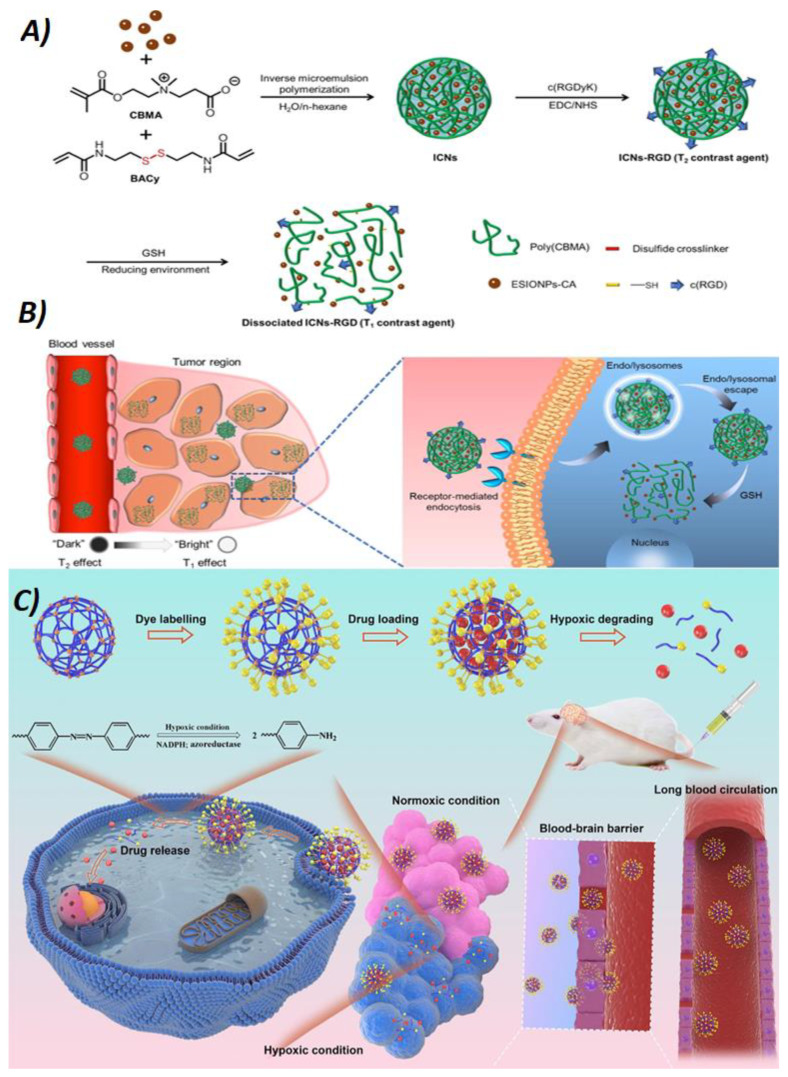
(**A**) Scheme for the preparation of poly(carboxybetaine methacrylate) (CBMA) nanogels (NGs) crosslinked with N,N-bis(acryloyl)cystamine(BACy), loaded with extremely small iron oxide nanoparticles (ESIONP), and modified on the surface with c(RGD) ligands, which bound α_v_β_3_ integrins, overexpressed on the membrane of some tumor cells. These NGs (ICNs-RGD) act as activatable MRI contrast agents with a switchable function, from a T2 contrast agent to a T1 one, through the stimuli-responsiveness toward GSH. (**B**) Depiction of the utilization of ICNs-RGD to realize a precise tumor diagnosis based on the transformation from the T2 to T1 contrasting effect at the tumor region. (**C**) Schematic illustration of the poly(phosphorylcholine)-based (HPMPC) NGs, which have long blood circulation, blood–brain barrier (BBB) penetration capability, and hypoxic-controlled drug release for glioblastoma drug delivery. In blue: phosphorylcholine polymers; in orange: azobenzene-containing crosslinker (molecule shown on the left); in yellow: Cy5 dye with a linker; in red: doxorubicin. Panels **A** and **B** reprinted from Ref. [125] with permission from copyright © 2020, American Chemical Society. Panel **C** reprinted with permission from Ref. [119].

**Table 1 nanomaterials-14-01300-t001:** Advantages and disadvantages of natural and synthetic polymers used in NGs.

**Natural Polymers**	
**Advantages**	**Disadvantages**
Biocompatibility and biodegradability Potential therapeutic effect (i.e., antimicrobial properties) Renewable and sustainable sources Potential for biofunctionalization Ability to retain water very efficiently	Batch-to-batch variability Limited mechanical strength Potential complex purification processes
**Synthetic Polymers**	
**Advantages**	**Disadvantages**
Controlled and reproducible properties Tailorable degradation rates High mechanical strength Versatile chemical modification/derivatization	Potential toxicity due to non-biodegradable synthetic residues Possible environmental impact Limited bioactivity Difficult and expensive synthesis procedure

**Table 2 nanomaterials-14-01300-t002:** Advantages and disadvantages of physical methods and chemical methods in batch synthesis of nanogels (NGs).

**Physical methods**	
**Advantages**	**Disadvantages**
Simple, rapid, and efficient synthetic processes Self-assembly and versatile (easily tunable size) structures No crosslinking agents or polymerization initiators, thus generally high safety and biocompatibility Potential high encapsulation of charged or polar drugs or biomolecules Suitable for large-scale production For amphiphilic NGs: high thermophysical and structural stability and high and precise control over loading and release profiles, especially of poorly water-soluble payloads.	Possible limited control over particle size and distribution Possible degradation of sensitive biomolecules Equipment can be expensive and complex Risk of contamination
**Chemical methods**	
**Advantages**	**Disadvantages**
Precise control over particle size and distribution Highly selective, reliable, versatile, and efficient synthetic processes Suitable for a wide range of materials Simple to functionalize NGs with various chemical groups Both hydrophobic and hydrophilic drugs can be encapsulated Suitable for large-scale production Short reaction times, high yield, and purity of products (with click reactions) Orthogonal, highly reactive, high yields (with click-like reactions) Environmentally friendly (in some cases) Can produce highly porous NGs	Use of potentially toxic reagents (e.g., cytotoxicity of copper for copper-catalyzed click-reactions, CuAAC) Complex purification processes Longer synthesis times compared to physical methods Possible necessity of surfactants that can be challenging to remove May require expensive reagents and catalysts

**Table 3 nanomaterials-14-01300-t003:** Comparison between batch synthesis and microfluidic synthesis, with respective benefits and drawbacks in chemical production and research.

**Batch Synthesis**	
**Advantages**	**Disadvantages**
Simple and common procedures Easier troubleshooting Potentially low initial setup cost Suitable for small-scale production	Generally limited control over reaction conditions Potential long reaction times Batch-to-batch variability Scalability issues
**Microfluidic Synthesis**	
**Advantages**	**Disadvantages**
Continuous process that can be automated Better heat and mass transfer, thus shorter reaction times and higher throughput Enhanced control over reaction conditions Safer reactions High scalability and reproducibility; suitable for large-scale production Environmentally friendly	Higher initial setup cost May require specialized equipment and expertise in designing and optimizing the flow system Potential clogging issues More complex troubleshooting

## Data Availability

Data are contained within the article.

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
