# Peer review of "Nanogels: Recent Advances in Synthesis and Biomedical Applications"

_nanomaterials, 2024, doi:10.3390/nano14151300_

Round 1

Reviewer 1 Report

Comments and Suggestions for Authors

This review presents the synthesis and biomedical applications progress of nanogels, which will accelerate the development of nanogels in biomedical applications. The whole manuscript was present in a relative good organizing, but in a bad writing.

(1) Extensive editing of English language required.

(2) As nanogels has been defined as NGs at line 28, the abbreviation NGs should used in the whole manuscript.

(3) The formation of references cited in the manuscript should revised.

(4) For the second paragraph of Introduction, the NGs can be composed of natural or synthetic polymers, as well as (synthetic polymer) chemical modified natural polymer, which should also introduced and discussed.

(5) A table about the advantages and disadvantages of natural or synthetic polymers used in NGs should provided after the second paragraph.

(6) Other reviews about NGs should also been cited and discussed in the Introduction.

(7) In the section of Batch Synthesis, the advantages and disadvantages of physical methods and chemical methods should compared in a table.

(8) Does section 2.1.2 Amphiphilc Properties is one of 2.1 Physical methods?

(9) The advantages and disadvantages of 2. Batch Synthesis and 3. Flow Chemistry Synthesis should compared in a table.

(10) The title of section 4. Stimuli Responsive Nanogel is not suitable, which should be the Properties of NGs, such as 4.1, 4.2, .....

(11) In the Conclusion, the issues and future direction of NGs should summarized point-by-point.

(12) The format of References is terrible. Eg: Ref.34, the doi is confused. Ref.124, the page number is missing. Rer.135, only author and title. The format of References should revised carefully.

Comments on the Quality of English Language

Extensive editing of English language required

Author Response

Thank you for reading our work and providing us with your precious feedback. We improved the manuscript thanks to the suggestions of yours and of the other reviewer, and we are confident that in the current form you would consider it acceptable for publication in Nanomaterials. Please find the detailed responses below and the corresponding revisions highlighted with the “track changes” tool of Word in the submitted pdf files (one with and one without changes in the references shown).

Comment 1: Extensive editing of English language required.

Response 1: We carefully proofread the manuscript, making some corrections especially in the introduction and in the conclusion, and paying attention to use American English consistently.

Comment 2: As nanogels has been defined as NGs at line 28, the abbreviation NGs should used in the whole manuscript.

Response 2:  Thank you for pointing this out. We now use “NG” or “NGs” almost everywhere, except at the beginning of paragraphs, in the section titles, and in the captions (where it is redefined if necessary).

Comments 3: The formation of references cited in the manuscript should revised.

Response 3: We are sorry we did not use the right formatting for references. We have now carefully checked all the inserted references, and formatted them according to the Nanomaterials/MDPI standard using “Mendeley”.

Comments 4: For the second paragraph of Introduction, the NGs can be composed of natural or synthetic polymers, as well as (synthetic polymer) chemical modified natural polymer, which should also introduced and discussed.

Response 4: Thank you for your comment. However, we humbly note that Section 1 already addressed this distinction, as modified natural polymers are included in the category of natural polymers. Furthermore, in the Section 2, about synthesis, the possible modifications of natural polymers have been discussed. Therefore, we only added a short sentence in the Introduction about the possibility of natural polymers modification, adding a reference and citing section 2.

Comments 5: A table about the advantages and disadvantages of natural or synthetic polymers used in NGs should provided after the second paragraph.

Response 5: We thank the Reviewer for his suggestion of adding this and other tables, which help in a review. We have now included the suggested table as Table 1.

Comments 6: Other reviews about NGs should also been cited and discussed in the Introduction.

Response 6: Many of the citations in the Introductions are indeed reviews on nanogels from different perspectives. We made this explicit in one of the final sentences of the first section, but we believe that doing so everywhere would only make the text harder to read. Moreover, the contents of other reviews were already explicitly cited in the most appropriate sections, in particular in the ones on synthesis and on applications. In any case, we also cited some additional reviews in the current version of the Manuscript.

Comments 7: In the section of Batch Synthesis, the advantages and disadvantages of physical methods and chemical methods should compared in a table.

Response 7: We have included the suggested table as the new Table 2.

Comments 8: Does section 2.1.2 Amphiphilc Properties is one of 2.1 Physical methods?

Response 8: Amphiphilic nanogels are counted among the nanogels cross-linked by physical methods because they do not require the use of cross-linking agents making covalent bonds, as discussed in detail in the text. In order to avoid any confusion, we changed the title of subsection 2.1.2 to “Hydrophobic Interactions in Amphiphilic Nanogels”

Comments 9: The advantages and disadvantages of 2. Batch Synthesis and 3. Flow Chemistry Synthesis should compared in a table.

Response 9: we have included the suggested table as the new Table 3.

Comments 10: The title of section 4. Stimuli Responsive Nanogel is not suitable, which should be the Properties of NGs, such as 4.1, 4.2, .....

Response 10: We are sorry, but we are not sure we understand correctly this comment… In any case, we considered important to add a section (the cited one) on stimuli-responsive nanogels, because of their broad application especially in drug delivery. We believe therefore that changing the title would confuse the readers. This section is quite short, because there are already recent (cited) reviews that discussed this topic in details, as already stated in the Manuscript; for this reason, we believe that dividing the section in subsections is not feasible. Properties of nanogels are discussed (also) in the other sections.

Comment 11: In the Conclusion, the issues and future direction of NGs should summarized point-by-point.

Response 11: We agree with the Reviewer that sometimes a schematization “point by point” can help in understanding the content of a text; indeed, we added three tables where this schematization has been done. However, after trying to follow the reviewer’s suggestion, we liked more our stylistic choice for the “Conclusion” section, and we would rather keep it like this. Also because we would have repeated many concepts already expressed in the previous sections, if we followed the Reviewer suggestion. In any case, we made some changes to the mentioned Section, hoping to improve its readability.

Comment 12: The format of References is terrible. Eg: Ref.34, the doi is confused. Ref.124, the page number is missing. Rer.135, only author and title. The format of References should revised carefully.

Response 12: See Response 3. We made the necessary changes to the (previous) citations 124, 134 and 135 (now 134, 145 and 146). The doi of the previous citation 34 (now 38), even if strange, was and is correct.

Response to Comments on the Quality of English Language

Comment: Extensive editing of English language required

Response: See Response 1 above. We carefully checked all the text in the manuscript and changed the language of the text to English (United States)

Reviewer 2 Report

Comments and Suggestions for Authors

A carefully designed and very good written comprehensive review concerning synthesis and potential use of nanogels as per current state of art.

Several comments or rather suggestions are listed below

Is there a correlation between method of synthesis and the potential use of nanogel formulation in the specific areas described or in others?  Does the method of synthesis orients/influence the modality of interaction with living matter and if so, what would be the most appropriate modality to choose a certain method of fabrication depending on the application desired?  Is it possible to introduce one two phrases regarding current applications of the different synthesis methods described?

Maybe introducing several modalities  in which nanogel are characterized  in terms of physical and chemical properties would be beneficial. What are the methods/equipment used for determining viscosity, rheology pore size (if the case) that need to be known before embarking in the design of a certain application in biology?

Maybe the authors would consider a brief description of magnetic nanogels that add magnetic responsiveness for the purpose of drug delivery, tracking systems and call fate modulation (10.15419/bmrat.v10i1.789, 10.1002/btm2.10190)

Regarding biomedical application potential, what is the perspective for large scale production /manufacturing of such materials and how does the upscaling affects batch reproducibility ?

Biocompatibility and safety subchapter reads rather general. Indeed, we all know we need to ascertain biocompatibility and safety but what are the practical modalities to do so? Please take into account that biocompatibility and safety needs to be designed and very well described from the very first stage of in vitro product testing. Doing this exclusively in the last phase of clinical trials is not only costly but bluntly dangerous and consistently belittles the chances of clinical trial authorisation. Are there methods to test in vitro and in vivo (animal model) biocompatibility, biodistribution) that are specific to nanogels?

Author Response

We are happy that you appreciated our manuscript, and we thank you for your suggestions that helped in improving it. Please find the detailed responses below and the corresponding revisions highlighted with the “track changes” tool of Word in the submitted pdf files (one with and one without changes in the references shown).

Comments 1: Is there a correlation between method of synthesis and the potential use of nanogel formulation in the specific areas described or in others?  Does the method of synthesis orients/influence the modality of interaction with living matter and if so, what would be the most appropriate modality to choose a certain method of fabrication depending on the application desired?  Is it possible to introduce one two phrases regarding current applications of the different synthesis methods described?

Response 1: Thank you for this insight. However, we cannot make a definitive statement on this matter. Of course, in clinical applications one should use biocompatible, better if natural, polymers, and try to avoid toxic reagents in the synthesis of the nanogels, and this is now better highlighted in the manuscript, in particular in the added tables and in the additions in the last two sections. However, concerning the synthesis modality, multiple methods can yield similar results, leading to nanogels with comparable properties and uses. We added a sentence about this in the introduction of section 5, and there already were some examples of applications in Sections 2 and 3, following the descriptions of the syntheses.

Comments 2: Maybe introducing several modalities  in which nanogel are characterized  in terms of physical and chemical properties would be beneficial. What are the methods/equipment used for determining viscosity, rheology pore size (if the case) that need to be known before embarking in the design of a certain application in biology?

Response 2: Thank you for the suggestion. A section on the characterization techniques for nanogels has been added to Section 6 in the resubmitted manuscript.

Comments 3: Maybe the authors would consider a brief description of magnetic nanogels that add magnetic responsiveness for the purpose of drug delivery, tracking systems and call fate modulation (10.15419/bmrat.v10i1.789, 10.1002/btm2.10190)

Response 3: We have added a section on magnetic nanogels in Section 4, including the citations you suggested and two additional ones.

Comments 4: Regarding biomedical application potential, what is the perspective for large scale production /manufacturing of such materials and how does the upscaling affects batch reproducibility ?

Response 4: This is a very interesting topic to discuss; we have now highlighted even better that flow chemistry could help for both upscaling and commercial batch reproducibility. However, as noted in the manuscript (especially in the conclusions) there are limited data available for making such inferences. For this reason, we felt it was not appropriate to speculate on this matter further than necessary.

Comments 5: Biocompatibility and safety subchapter reads rather general. Indeed, we all know we need to ascertain biocompatibility and safety but what are the practical modalities to do so? Please take into account that biocompatibility and safety needs to be designed and very well described from the very first stage of in vitro product testing. Doing this exclusively in the last phase of clinical trials is not only costly but bluntly dangerous and consistently belittles the chances of clinical trial authorisation. Are there methods to test in vitro and in vivo (animal model) biocompatibility, biodistribution) that are specific to nanogels?

Response 5: We completely agree with what you write, and we took the liberty to adapt your sentence about early biosafety tests in an addition in Section 6 of the manuscript. We acknowledge that these topics are covered in a general manner. However, there is currently no standard for preclinical studies of nanogels or, more broadly, nanostructured formulations, as they are designed on a case-by-case basis. Furthermore, there are few validation projects underway, so there are no specific tests we can cite as examples. In the new version of the manuscript, we expanded section 6, hoping to clarify it and to avoid any confusion for the readers about this topic.

Round 2

Reviewer 1 Report

Comments and Suggestions for Authors

The paper was corrected taking into account all my comments. 

Comments on the Quality of English Language

Moderate editing of English language required